# Idiopathic Infertility as a Feature of Genome Instability

**DOI:** 10.3390/life11070628

**Published:** 2021-06-29

**Authors:** Agrita Puzuka, Baiba Alksere, Linda Gailite, Juris Erenpreiss

**Affiliations:** 1Scientific Laboratory of Molecular Genetics, Riga Stradins University, LV-1007 Riga, Latvia; agrita.puzuka@rsu.lv (A.P.); baiba.alksere@rsu.lv (B.A.); linda.gailite@rsu.lv (L.G.); 2Department of Biology and Microbiology, Riga Stradins University, LV-1007 Riga, Latvia; 3EAA Andrology Centre, Clinic IVF-Riga, LV-1010 Riga, Latvia

**Keywords:** male infertility, DNA breaks, genome instability

## Abstract

Genome instability may play a role in severe cases of male infertility, with disrupted spermatogenesis being just one manifestation of decreased general health and increased morbidity. Here, we review the data on the association of male infertility with genetic, epigenetic, and environmental alterations, the causes and consequences, and the methods for assessment of genome instability. Male infertility research has provided evidence that spermatogenic defects are often not limited to testicular dysfunction. An increased incidence of urogenital disorders and several types of cancer, as well as overall reduced health (manifested by decreased life expectancy and increased morbidity) have been reported in infertile men. The pathophysiological link between decreased life expectancy and male infertility supports the notion of male infertility being a systemic rather than an isolated condition. It is driven by the accumulation of DNA strand breaks and premature cellular senescence. We have presented extensive data supporting the notion that genome instability can lead to severe male infertility termed “idiopathic oligo-astheno-teratozoospermia.” We have detailed that genome instability in men with oligo-astheno-teratozoospermia (OAT) might depend on several genetic and epigenetic factors such as chromosomal heterogeneity, aneuploidy, micronucleation, dynamic mutations, RT, PIWI/piRNA regulatory pathway, pathogenic allelic variants in repair system genes, DNA methylation, environmental aspects, and lifestyle factors.

## 1. Introduction

Infertility (defined as the inability to conceive within a year of unprotected coitus) is a global health and social issue affecting close to 15% of couples. In half of the couples seeking medical treatment for infertility, male factor infertility is identified together with abnormal semen characteristics [1].

Increasing infertility is considered to be mostly dependent on the lower fecundity of women planning their first pregnancy at a more advanced age, coupled with decreasing semen quality in men [2]. In 1992, it was reported that semen quality had markedly deteriorated over the previous 50 years [3]; this finding has been subsequently confirmed [4]. The reasons for such a decline are still not clear. A possible interplay of genetic, environmental, and lifestyle features has been suggested. However, our knowledge of the impact proportions is presently extremely limited, and a significant number of male infertility cases and decreased sperm quality remain unexplained. Current insufficient understanding of sperm production (spermatogenesis) biology limits the possibility of a precise diagnosis in up to 75% of severe male infertility cases where no causation factor is detected [2]. These cases are termed ‘idiopathic male infertility’.

The focus of current clinical practice is laid on the presence of sufficient sperm count in the ejaculated specimen with adequate motility and morphology of spermatozoa capable of giving fertilization a chance (conventional semen analysis). However, semen analysis fails to predict the fertilizing potential of the male gamete precisely. Indeed, a substantial overlap of semen parameters between fertile and infertile males has been reported [5]. Moreover, the semen analysis ability to indicate the appropriate artificial reproductive techniques for couples diagnosed with severe or unexplained infertility also appears to be limited [6,7]. For these very reasons, researchers started exploring possible genetic disorders behind male infertility as well as the utilization of additional tests to gain more insight into the reproductive capacity of an individual [8]. An extensive amount of male infertility research has been dedicated to the investigation of the role played by sperm DNA integrity (secondary DNA structure related to chromatin packaging). Evaluation of sperm DNA integrity has proved to be a strong predictor of male infertility in vivo, providing an independent marker only partly related to the conventional semen parameters [9,10,11,12,13].

An abnormal sperm chromatin structure (defective histone-to-protamine replacement) has been found to be associated with the incomplete maturation of sperm during spermiogenesis [14]. However, other causative mechanisms, including abortive apoptosis and oxidative stress, have also been described [15]. A systematic review and meta-analysis have concluded that DNA-damaged spermatozoa will affect the outcome of conventional IVF but are unlikely to influence the results of ICSI treatment [15,16,17]. Nevertheless, with different groups presenting different results (perhaps due to variations between clinics with respect to patient proportions with different genetic backgrounds and methodology (based on the staining equilibrium principle, reviewed in [15,18,19,20]), this remains a topic of debate.

The central biological event of fertilization is the transition of the paternal and maternal genomes to the offspring. Genome instability—understood as reduced fidelity with which genetic information is passed on to daughter cells—can impair both mitosis and meiosis. Changes taking place in the course of DNA replication, DNA repair, and cell division (chromosome duplication, recombination, segregation) provide a natural source of genome instability. Depending on the nature of processes involved, genome instability can lead to (a) single nucleotide variants and microsatellite expansions caused by errors in DNA synthesis and defective DNA repair mechanisms; (b) variations in the number of chromosomes (aneuploidy) due to mistakes of the chromosome segregation apparatus; (c) other types of genetic modifications, such as gross chromosomal rearrangements, copy number variants (CNVs), hyper-recombination and loss of heterozygosity, triggered in the majority of cases by single-strand DNA gaps or double-strand breaks (DSBs) occurring as a consequence of replication stress [21]. Apart from genetic variations, growing attention is now being paid to epigenetic factors that influence gene expression but are not caused by alterations in the DNA sequence. Said epigenetic variations are based on DNA methylation, post-translational modifications of histone tails, and non-coding RNA molecules.

It was recently suggested that genome instability could lead to decreased general health and increased morbidity [22]. It is possible that infertility or decreased semen quality could be among the drawbacks or symptoms of genome instability. Somatic chromosome mosaicism and chromosome instability are likely mechanisms or elements of the pathogenetic cascade of a wide spectrum of pathologies and can mediate inter-individual genetic variation, prenatal development, and aging [23]. Genome instability may be responsible for severe cases of infertility in men—a multifactorial condition manifested by disrupted spermatogenesis, among other traits. Research in male infertility has provided evidence supporting the assumption that impaired spermatogenesis (particularly in idiopathic infertility cases) is often associated with factors not limited to testicular dysfunction. An increase in the frequency of urogenital disorders and rising incidence rates of various types of cancer (testicular, prostate, colon, melanoma), as well as a decline in overall health (manifesting itself through lower life expectancy and higher morbidity), have been reported in infertile men [24,25]. The pathophysiological link between lower life expectancy and male infertility supports understanding of the male infertility phenomenon as a systemic rather than an isolated condition [26].

An array of genetic and epigenetic factors of male infertility, including single nucleotide variants, CNVs, protamine content, methylation characteristics, protein content, and small RNAs, have been analyzed in various studies. Currently, the above assays are only used within the research setting being not routinely carried out by fertility clinics [27]. In the present review article, we survey the data on the association of male infertility with genetic alterations, their respective causative factors, and consequences, as well as the methods for genome instability assessment.

The content of the review is schematically summarized in Figure 1.

## 2. Methodology

A literature search in Pubmed and Google Scholar databases for MeSH terms referred to genome instability, male infertility, genetic variations, epigenetic regulation, repair system, senescence, and environmental agents was performed until October 2020. The most important results were further described in this review article.

## 3. Reasons and Consequences of Genome Instability

### 3.1. Genetic Variations

In this section, well-established genome changes associated with reduced male fertility are reviewed, beginning with genome abnormalities visible by microscope (cytogenetic variants), followed by microdeletions (azoospermia factor (AZF) region variants) and microsatellite instability (MSI), then finally, single-gene allelic variants. Appendix A in Appendix A summarizes the latter data, including the normal function and pathological dysfunction of these genes.

#### 3.1.1. Aneuploidy

Aneuploidy (abnormal chromosome number) is the most prevalent cause and consequence of genome instability. The most common chromosomal aneuploidy in azoospermic patients is Klinefelter syndrome (47,XXY), accounting for approximately 11% of azoospermia cases [28]. Up to 40% of spermatozoa show aneuploidy in men carrying Robertsonian translocations, occurring in 1:813 neonates [29]. Reciprocal translocations are found even more often (1:712 neonates), and the proportion of spermatozoa with chromosome aberrations in this group varies between 19–81% [29,30]. A substantial proportion of genetic factors in male infertility is linked to postnatal alterations that can occur either in all tissues or only in reproductive organs and spermatozoa (gonadal mosaicism). Separately, each of these alterations is rare, but because there are many genes involved in spermatogenesis, it is possible that the real proportion of genetic factors in male infertility is much higher [31].

#### 3.1.2. Copy Number Variants (CNVs)

Another cause of genome instability is CNVs—a phenomenon characterized by the presence of repeated genome sections with the number of repeats in the genome varying between individuals. A classic and simple example of CNVs related to male infertility is **microdeletions** in the AZF locus of the Y chromosome [32,33,34]. The nature of the Y chromosome AZF regions, rich in repetitive sequences, makes it prone to frequent rearrangements, often leading to full or partial AZF deletions [35,36]. AZF deletions are found to be the causative factor in nearly 7% of afflicted men Worldwide, making these deletions the most common known genetic cause of impaired spermatogenesis [36,37,38].

The frequency of Yq microdeletions in different parts of the world varies slightly. This could be explained by differences in sample size, methodology used, and the population screened. The lowest prevalence of Yq microdeletions is estimated in Europe (3%) and Australia (5%), while the rest of the world has an average of 8–9% [39].

#### 3.1.3. Microsatellite Instability (MSI)

Genome instability can be caused by MSI, when the number of repeated DNA bases in a microsatellite differs from what it used to be when the microsatellite was inherited. MSI represents a state of genetic hypermutability (predisposition to mutation) resulting from impaired DNA mismatch repair (MMR) [40]. The dynamic mutation of microsatellite trinucleotide repeats is believed to be triggered by errors in germline cell division cycles. These frequently mutating loci can be used as markers of genetic instability throughout a cell cycle [41]. MSI may sometimes occur as a result of epigenetic silencing of MMR genes [26].

A variable MSI-related region is found in the androgen receptor (*AR*) gene [42,43]. The AR is a hormonal transcription factor mediating the physiologic and pathophysiologic effects of androgens, such as sexual differentiation and prostate development. The AR also impacts cancer progression by binding to genomic androgen response elements, which affect the transcription of AR target genes [44]. The *AR* gene contains a region with variable numbers of copies of the trinucleotides CAG and GGN. Increased copy numbers of CAG and GGN are associated with impaired spermatogenesis [26,43]. However, evidence to support this has yet to be presented. Genome instability in general and, more specifically, MSI is associated with increased cancer risk. Accordingly, short CAG and GGN repeats in the *AR* gene have been found to be correlated with a higher risk of prostate cancer, especially in Caucasian men [45].

#### 3.1.4. Single-Gene Allelic Variants

It is important to mention that there may emerge single-gene allelic variants, such as pathogenic variants in *CFTR* gene, displaying a strong association with azoospermia. The protein this gene encodes operates as a channel across the membrane of cells with exocrine function, thereby besides pulmonary and digestive system dysfunction affecting the reproductive function as well. In recent years, several other allelic variants have been reported to have a strong association with non-obstructive azoospermia (e.g., changes in the *TEX11* gene, an X-linked gene associated with crossing over in meiosis [46]) and severe forms of teratozoospermia (changes in *AURKC*, *DPY19L2*, *DNAH1,* and other genes [47,48]).

#### 3.1.5. Activation of Retrotransposons

Mobile genetic elements in humans have been categorized as DNA transposons and retrotransposons (RT). DNA transposons move driven by a cut-and-paste mechanism, whereas the process of retrotransposition involves RT mobilization by a copy-and-paste mechanism via an RNA intermediate [49]. DNA transposons account for 3% of the human genome [50]. As DNA transposons are found to be inactive in humans [50,51], they are not reviewed further in this article.

RT are abundant in all types of organisms, comprising 45–50% of the human genome. RT are the most widespread class of transposons in mammals and represent a potential source of genome instability, mutations, and polymorphisms [52,53]. RT drive mutations in two ways: (1) by jumping to new locations; (2) by inducing chromosomal aberrations via non-allelic homologous recombination [54]. Abnormally excessive RT activity is associated with different pathologies–cancer, developmental disturbances and other diseases [55].

The silencing of RT in the germline is a unifying mechanism of PIWI-interacting RNAs (piRNAs) observed in all animal species. Moreover, piRNAs have recently been proven essential for successful spermatogenesis controlling transcription [56].

### 3.2. Repair System Defects, Genome Instability and Male Infertility

#### 3.2.1. Replication Errors

The average mutation rate of single base substitutions in humans per generation has been calculated to reach ~1–1.5 × 10^−8^. Each additional year of the father’s age at conception has been found to increase the germline mutational load of the offspring by one to three de novo mutations. Due to the fact that said process involves recurrent rounds of DNA replication and cell division, random copy-error mutational events are expected to occur mainly in the male germline [57].

#### 3.2.2. Mismatch Repair Genes

Repair of damaged DNA occurs only in early spermatogenesis. The oocyte is capable of repairing sperm DNA to maintain sperm/embryo genomic integrity and stability [58,59]. One of the DNA repair systems correcting mispaired bases during DNA replication errors is mismatch repair (MMR). Allelic variants in MMR *MSH* genes can be responsible for the MMR failure in sperm DNA and male infertility [60,61,62]. The rs26279 variant in *MSH3* is associated with idiopathic male infertility, which makes it a possible risk factor for idiopathic male infertility [62]. The *MSH3* gene located at 5q11-13 encodes an MSH3 protein playing an essential role in the MMR system. MSH3 also recruits other MMR proteins to the base mismatch loci to fix the DNA damage [63]. In addition, rs1800734 and rs4647269 in the *MLH1* gene showed an association with severe oligozoospermia [62]. *MLH1* promoter is significantly more methylated in oligozoospermic patients. Furthermore, a statistically significant positive association was observed between elevated reactive oxygen species (ROS) levels and *MLH1* methylation, with no such association found in the case of *MSH2* methylation [61].

#### 3.2.3. Defects of DNA Mitotic and Meiotic Repair Genes

The most deleterious type of DNA damage is breaks in DNA double-strand (DSBs). They can occur endogenously by virtue of DNA transcription and replication through the impaired activity of topoisomerases I and II [64,65] or exogenously [66]. DSBs arise naturally during meiotic recombination, as this process requires breaking to be endogenously induced and then followed by ligation of DNA molecules by topoisomerase-II-like enzyme (SPO11). Uncorrected or misrepaired DNA DSBs may result in genome instability and cause genetic aberrations, apoptosis, and infertility [60]. Non-homologous end-joining or homologous recombination are the two mechanisms that can be involved in DSBs repair. While in meiotic prophase, SPO11 generates several hundred programmed DSBs, which are exclusively repaired by homologous recombination, promoting obligate crossover between homologous chromosomes (at least one pair). The unique challenge of X-Y recombination for male meiosis is that a very small homology segment (so-called “pseudo-autosomal region”) on the tip of the X-chromosome arm is responsible for it [67]. It is the most vulnerable region in the male genome [68]. MRE11-RAD50-NBS1 (MRN) complex-dependent DNA end resection is a prerequisite for homologous recombination repair in somatic cells. After being generated, all meiotic DSBs remain linked with SPO11; however, the MRN complex is also essential for the repair of meiotic SPO11-linked DSBs in male mice [69]. SPO11 knockout activates abnormal meiotic arrest in zygotene spermatocytes in rats, causing male infertility [70]. Spo11P306T/− mice are sterile; also, they make fewer meiotic DSBs than Spo11+/− animals, which suggests that the Spo11P306T allele is possibly delayed in making sufficient DSBs at the right time. If consequences are the same in humans, phenotypes of premature ovarian failure, low sperm counts, as well as a probable increase in the numbers of aneuploid gametes could be predicted [71].

Oxidative stress directly damages guanine residues in DNA. Even though this process is initially reversible, it is capable of impairing DNA double-strand architecture and ultimately inducing single-strand breaks (SSBs). Male germ cells are not equipped with molecular SSB repair mechanisms, but nevertheless, at the spermatid stage, sperm can rely on the base excision repair capability for the removal of oxidative stress-induced transversion mutations. In this mechanism, altered DNA bases are identified, and an abasic site is generated, the next step being cleavage by an endonuclease. After zygote formation, paternal SSBs and DSBs can be effectively repaired by the oocyte when the DNA damage in sperm is below 8% [59]. Male partners with SSBs in <25% of their sperm have an average live birth rate of 33% following IVF, while those with SSBs in >50% of their sperm showed a markedly reduced live birth rate of 13% [59]. Furthermore, a correlation between the presence of SSBs in sperm and increased risk of miscarriage associated with a male factor has been observed [72].

### 3.3. Epigenetic Changes and Male Infertility

#### 3.3.1. DNA Methylation

During embryonic development, the mammalian genome is exposed to two waves of global demethylation and remethylation. The first wave occurring after fertilization includes the erasure of most methylation marks inherited from the gametes, followed by the establishment of the embryonic methylation pattern with the onset of differentiation determination [73]. Developmental reprogramming opens a risky escape route for transposable elements from their hypermethylated silencing shelters needed for reprogramming in the cleavage embryo [74]. The second wave occurring in the germline is initiated upon the erasure of global methylation in primordial germ cells (PGCs) in the genital ridge and completed following the establishment of sex-specific methylation patterns during later stages of germ cell development in the gonad. Replacement of histones by protamines in spermiogenesis is also associated with genome instability, which provides the environment for the induction of de novo mutations [75].

#### 3.3.2. Histone Modifications

Spermatogenesis is a set of processes setting into motion in the postembryonic period of ontogenesis with the appearance of primordial germ cells (PGCs) that are destined to migrate to and populate the genital ridge. PGCs undergo various epigenetic modifications that could be further subdivided into two events—global erasure and paternal reestablishment of the imprinted regions. Male PGCs undergo several mitotic division rounds in the embryonic testes until they enter mitotic arrest (at this point, ~25,000 are found) [56].

The role of histones in spermatogenesis has been widely considered in the literature. There are two basic histone subtypes, so-called canonical histones, that are involved in genome packaging and gene regulation, and non-canonical histones (histone variants) regulating the initiation of transcription DNA repair. In mammals, 11 histone H1 subtypes are known, with H1t, H1T2, HILS1 being expressed specifically in the testis. The testis-specific H1T2 variant expression is observed in haploid male germ cells until histone-to-protamine transition [76]. Loss of H1T2 affects nuclear condensation and highly reduces male fertility [77].

#### 3.3.3. PIWI/piRNA Pathway

The PIWI/piRNA pathway is recognized as a major mechanism protecting the germ cell genome from insertional mutagenesis by RT [78]. The germline genome is assumed to be protected via paternal small non-coding miRNAs, siRNAs, piRNAs, qRNAs, and DNA satellite-associated non-coding RNA mechanisms. Coding and non-coding parental RNAs are to be of adequate quality and quantity to secure embryo growth. Small non-coding RNAs are short (<100 nucleotides in length), abundant key regulators of cellular mechanisms. Most sperm small non-coding RNAs fall into four major classes: repetitive elements, transcription start site/promoter-associated, piRNAs, and miRNAs. During spermatogenesis, the role of piRNAs is to suppress the activation of mobile transposable elements. The absence of these regulatory RNAs can lead to spermatogenic arrest. They are capable of protecting genome integrity as they bind to DNA and thus prevent the impact of various classes of repetitive and transposable elements such as SINE, LINE, MER, and LTR at specific stages of embryogenesis [79].

Failure to suppress RT during spermatogenesis often results in DSBs in sperm DNA. Therefore, it is important to identify pathogenic allelic variants in repair system genes. Allelic variants in the two DNA recombinases RAD51 and DMC1—indispensable in meiosis mediating DSB repair system genes and DNA MMR genes in the case of MSI—could be important as they play decisive roles in key pathways of DNA repair. Furthermore, given these genes are crucially involved in mitotic DNA repair and meiotic recombination, also being highly expressed in the testis, any decrease of the activity of their encoded protein may disrupt the DNA repair system and induce the accumulation of mutations and breaks in sperm DNA. Accordingly, variants in these MMR genes may affect sperm quality and reduce fertility [80].

### 3.4. Aging and Cellular Senescence

Aging is an ontogenesis phase characterized by the deterioration over a period of time of an organism’s physiological functions that are necessary for survival [81]. In the retrospective study encompassing in total 25,445 men treated at infertility clinics and 87 healthy men without reproductive problems, the level of DNA fragmentation was measured using sperm chromatin structure assay. It was found that advancing paternal age was linked with increased sperm DNA fragmentation scored as an increased percentage of sperm in semen ejaculates with measurable DNA strand breaks [82]. This large study confirms the hypothesis proposed in numerous earlier studies that older men produce more sperm with DNA damage [83].

Male aging manifests itself in a higher occurrence of meiotic errors, chromatin fragmentation, telomere shortening, epigenetic changes, and impaired control of mobile elements. The major biological markers of genome stability are telomeres with their sheltering proteins protecting chromosomes against DNA damage and ensuring correct chromosome alignment during DNA replication. Drastic telomere shortening is associated with decreased sperm parameters. DNA methylation and telomere integrity are also important for normal fertilization and embryo development [84].

Werner syndrome (WS) illustrates the link between cellular aging and male infertility. Besides features associated with premature aging (progressive skin atrophy, early graying and loss of hair, arteriosclerosis, diabetes mellitus, cancer), the phenotypic hallmark of Werner syndrome is male infertility due to hypogonadism leading to complete azoospermia [85]. The cause of WS is pathogenic variants (mostly loss of function variants according to the ClinVar database) in the WRN gene, which encodes a DNA helicase, a member of the RECQ family. The presence of an exonuclease domain in its N-terminal region of the protein makes WRN helicase unique. Biochemical and cell biological studies during the past decade have demonstrated involvements of the WRN protein in multiple DNA transactions, including DNA repair, recombination, replication, and transcription [85]. Since WS cells show prolonged S-phase and reduced frequency of DNA replication, that implicates the role of WRN helicase in DNA replication. This hypothesis is supported by the fact that WRN helicase interacts with several factors involved in DNA replication (RPA, PCNA, FEN-1, and Topoisomerase I). WS cells are hypersensitive to camptothecin-a Topoisomerase I inhibitor, which leads to accumulation of DNA damage, especially in the S-phase. Besides that, WRN helicase forms or unwinds the Holliday junction intermediate associated with a regressed replication fork. WS cells are hypersensitive to 4NQO, which induces oxidative damage. Additionally, WRN helicase is associated with other repair mechanisms-base excision repair (BER; interacts with BER factors, polδ, polβ, PCNA, RPA, FEN-1, and PARP-1), and double-strand break repair (interacts with the factors Ku, DNA-PKcs, and the Mre11-Rad50-Nbs1 complex, as well as the telomeric DNA protecting proteins, TRF1, TRF2, and POT1). Finally, findings also suggest that WRN helicase is involved in telomere metabolism since abnormal telomere dynamics in WS lymphoblastoid cell lines show weak or no telomerase activity. These DNA structures formed at telomere ends must be resolved during DNA replication to be accessible to DNA polymerases and telomerase, therefore, WRN helicase might function in the resolution of higher-order structures in telomeric DNA [86,87]. Besides the above-mentioned protein interactions, WRN interacts with the tumor suppressor protein p53 that is critical for regulation of the cell cycle arrest, apoptosis, and senescence. The ability of p53 to regulate WRN expression levels is supported by the finding that overexpression of p53 results in a decrease in transcription of the WRN gene [88].

At the end of the long arm of the human X chromosome encompassing the region from Xq27.2 to the q telomere, including the chromosomal band Xq28, there exists a fragile X-chromosome region of the DNA breakpoints, possibly due to the enrichment with LTR retrotransposons. This region is of particular interest since it contains the highest density of genes related to genetic diseases [89]. Subtelomeres constituting a safeguard for gene expression and chromosome homeostasis also play a particular role in taking care of telomere maintenance in case of their damage. In addition, all chromosomes contain at their subtelomeric region the species-specific sites for recognition and synapsis of homologous chromosomes in meiosis [90].

### 3.5. Environment and Lifestyle Factors and Their Influence on the Genome and Spermatogenesis

There is accumulating evidence that harmful environmental (phthalates, perfluoroalkyl compounds, etc.) and lifestyle factors (smoking, alcohol, mobile phones) impact DNA methylation in sperm and could consequently lead to male infertility [91]. It is unclear in what way the sperm epigenome is sensitive to potential mutagens and how these factors impact fertility parameters. Recent investigations have confirmed a link between potentially damaging agents and disruption to epigenetic processes [92,93], however, several studies have reported controversial results [94].

## 4. Assessment of Genome Instability

Maintenance of genome stability relies on the DNA damage response executed through a functional network consisting of signal transduction, cell cycle regulation, and DNA repair.

Various approaches can be used to investigate different types of genome instability in different tissues. These methods, such as sperm DNA fragmentation tests, micronucleation assays, or MSI investigation are successfully applied in clinical diagnostics and are further described in this article.

### 4.1. Assessment of Genome Instability in Sperm

Genome instability in sperm cells can be assessed by (1) sperm chromatin structural probes, (2) sperm nuclear matrix assays, (3) direct assessment of DNA strand breaks (reviewed in Erenpreiss et al., 2006), (4) ALU/LINE transcription. The processes inducing DNA damage in ejaculated spermatozoa are interrelated. Abnormal spermatid protamination and disulphide bridge formation as a result of inadequate thiols oxidation during epididymal transit, leads to diminished sperm chromatin packaging and makes sperm cells more susceptible to ROS-induced DNA fragmentation [15].

### 4.2. Evaluation of ALU/LINE Activity

Detailed investigation can determine transposon content and activity after stressful events [95]. A simple and convenient approach is to measure transposon expression by quantitative PCR using retroelement region-specific primers [96]. High-throughput sequencing can be modified to focus the investigation on transposable elements [97]. Long-read sequencing is a promising tool for obtaining more detailed transposon data [98]; however, alignment errors are still a challenge in bioinformatics [99].

### 4.3. Assessment of Genome Instability in Peripheral Blood Leukocytes

Various molecular genetics approaches can be employed to anticipate genome instability. Allelic variants and microdeletions in genes controlling MMR, RT, and other important mechanisms can be detected by Sanger sequencing, next-generation sequencing, and third-generation sequencing [100]. Historically, MSI was measured by capillary electrophoresis [101] but can now also be exposed by modern sequencing methods [102]. Expression analysis is a useful tool when investigating DNA damage. RNA sequencing is used to evaluate gene expression quantification, alternative splicing, differential expression, RNA editing, expressed variant identification, and co-expression networks [103].

A micronucleus is an erratic (third) nucleus that develops during the last step of cell division. As micronuclei contain fragments of chromosomes, their formation proceeds in a daughter cell with an aberrant chromosomal set. An increased number of micronuclei is a sign of chromosomal damage caused by genotoxic agents [104]. Micronucleus assays are widely used to measure genetic damage caused by mutagen exposure [105].

The COMET assay measures DNA strand breaks at the individual cell level, and it can be applied both in sperm (as described above) and peripheral blood cells. The fluorescence is calculated to measure the level of DNA damage [106]. The COMET assay is the only test that makes it possible to discern DNA SSBs from DNA DSBs [59].

## 5. Conclusions

In summary, we have presented extensive data supporting the notion that genome instability may lead to severe male infertility (OAT), whereas genetic tests recommended for routine clinical investigation (such as testing for karyotype, Y chromosome AZF region microdeletions) provide a diagnosis in only 15–25% of cases. We have detailed that genome instability in men with OAT depends on several other genetic and epigenetic factors such as chromosomal heterogeneity, aneuploidy, micronucleation, dynamic mutations, RT, PIWI/piRNA regulatory pathway, pathogenic allelic variants in repair system genes, DNA methylation, and environmental aspects (e.g., smoking, alcohol).

## Figures and Tables

**Figure 1 life-11-00628-f001:**
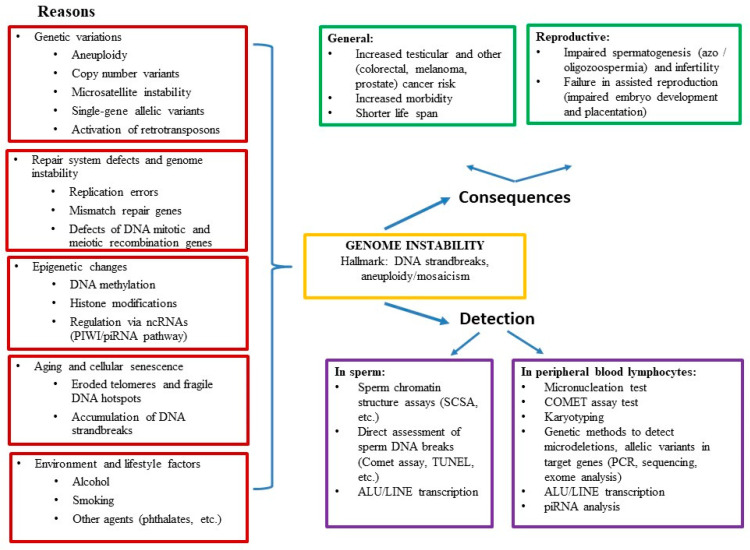
Structure of the article.

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
