# Peer review of "Idiopathic Infertility as a Feature of Genome Instability"

_life, 2021, doi:10.3390/life11070628_

Round 1
Reviewer 1 Report
The auther had detailed that genome instability in men with OAT depends on several other genetic and epigenetic factors including chromosomal heterogeneity, aneuploidy, micronucleation, dynamic mutations, RT, PIWI/piRNA regulatory pathway, pathogenic allelic variants in repair system genes, DNA methylation and environmental aspects. This was an well writing report and I approve of its publication. The literature review was comprehensive and justifies the authors' recommendation. However, I have a few additional comments:
1. The author stated "AZF deletions are found to be the causative factor in nearly 5% of afflicted men of European descent", in Asian ethnic groups, is there any difference in the proportion of infertility caused by AZF loss?
2. Previous study stated pathogenic variants in CFTR gene caused the CBAVD (obstructive azoospermia). However, this seems to be inconsistent with the type of infertility that the author would like to discuss.
3. Line 20, replace OAT with oligo-astheno-teratozoospermia (OAT)
Author Response
- The author stated "AZF deletions are found to be the causative factor in nearly 5% of afflicted men of European descent", in Asian ethnic groups, is there any difference in the proportion of infertility caused by AZF loss?Response1: There are some differences in the frequency of the Y chromosome microdeletion among the Word's populations, but these variations mainly are depending on patient inclusion criteria (more common Y microdeletions found in azoospermia patients then in severe oligozoospermia cases) and diagnostic accuracy of the laboratory. The frequency of the Y chromosome microdeletions in the article will be described more precisely.
- Previous study stated pathogenic variants in CFTR gene caused the CBAVD (obstructive azoospermia). However, this seems to be inconsistent with the type of infertility that the author would like to discuss. Response 2: Thank you for this remark. The info about CBAVD will be removed from the article since it is not the case of idiopathic male infertility, and info about CFTR pathogenic variants will be rephrased correspondingly.
- Line 20, replace OAT with oligo-astheno-teratozoospermia (OAT). Response 3: The abbreviation OAT in the line 20 will be replaced with full name of condition, respectively - oligo-astheno-teratozoospermia (OAT).
Reviewer 2 Report
Review of a manuscript entitled: “Idiopathic Infertility as a Feature of Genome Instability” by Agrita Puzuka and colleagues (life-1264348-peer-review-v1. https://doi.org/10.3390/xxxxx).
This manuscript reviews the evidence regarding association of male infertility with genetic, epigenetic and environmental factors, in particular genomic instability. The manuscript is well written and provides a thorough overview of the issue, but for one, surprising, omission. While the authors argue the complex interrelationships between genomic instability, senescence and male infertility the quintessential human genetic disorder embodying this triad, Werner syndrome, is not mentioned.
In addition to phenotypes related to premature ageing (e.g. progressive skin atrophy, early graying and loss of hair, arteriosclerosis, diabetes mellitus) the phenotypic hallmark of Werner syndrome is male infertility, due to complete azoospermia.
Cells from Werner syndrome patients show a prolongation and arrest in the S phase of the cell cycle, which results in a shortened proliferative lifespan (Poot M, Hoehn H, Rünger TM, Martin GM (1992) Impaired S-phase transit of Werner syndrome cells expressed in lymphoblastoid cell lines. Exp Cell Res 202: 267–273). In addition, Werner syndrome cells show a spontaneously elevated rate of deletions (Fukuchi K, Martin GM, Monnat RJ Jr (1989) Mutator phenotype of Werner syndrome is characterized by extensive deletions. Proc Natl Acad Sci USA 86: 5893–5897) and a defective response to DNA interstrand crosslinks and DNA double strand breaks (Poot M, Yom JS, Whang SH, Kato JT, Gollahon KA, Rabinovitch PS (2001) Werner syndrome cells
are sensitive to DNA cross-linking drugs. FASEB J 15: 1224–1226; Poot M, Jin X, Hill JP, Gollahon K A, Rabinovitch PS (2004) Distinct functions for WRN and TP53 in a shared pathway of cellular response to 1‐beta‐D‐arabinofuranosylcytosine and bleomycin. Exp Cell Res 296: 327–336). Finally the Werner syndrome protein functions as a molecular switch between the, more or less fatihful, canonical NHEJ and the error prone alt-NHEJ (Shamanna RA, Lu H, de Freitas JK, Tian J, Croteau DL, Bohr VA (2016) WRN regulates pathway choice between classical and alternative non-homologous end joining. Nat Commun 7:13785).
The information covered by these papers, and others, is highly relevant to the subject matter and should therefore be incorporated in this review.
Author Response
Werner syndrome, is not mentioned. In addition to phenotypes related to premature ageing (e.g. progressive skin atrophy, early graying and loss of hair, arteriosclerosis, diabetes mellitus) the phenotypic hallmark of Werner syndrome is male infertility, due to complete azoospermia.
Response: Thank you for the good remark. Werner syndrome, and this disease molecular mechanism and the link with male infertility will be included in this review.